# End-to-end Conditional Robust Optimization

**Abhilash Reddy Chenreddy**[1]                    **Erick Delage**[1]

[1]GERAD and Department of Decision Sciences, HEC Montréal, Canada

## Abstract

The field of Contextual Optimization (CO) integrates machine learning and optimization to solve decision making problems under uncertainty. Recently, a risk sensitive variant of CO, known as Conditional Robust Optimization (CRO), combines uncertainty quantification with robust optimization in order to promote safety and reliability in high stake applications. Exploiting modern differentiable optimization methods, we propose a novel end-to-end approach to train a CRO model in a way that accounts for both the empirical risk of the prescribed decisions and the quality of conditional coverage of the contextual uncertainty set that supports them. While guarantees of success for the latter objective are impossible to obtain from the point of view of conformal prediction theory, high quality conditional coverage is achieved empirically by ingeniously employing a logistic regression differentiable layer within the calculation of coverage quality in our training loss. We show that the proposed training algorithms produce decisions that outperform the traditional "estimate then optimize" approaches.

## 1   INTRODUCTION

In a standard machine learning setting, $\Psi \subseteq \mathbb{R}^m$ represent the input set and $\Xi \subseteq \mathbb{R}^m$ represent the output sets and we aim to learn a model $\mathfrak{F}_\theta$ parameterized by $\theta$ that approximates the relationship between the input and output by minimizing a loss function $\mathcal{L}$. In real-world applications, we usually have a dataset of $M$ samples, $\mathcal{D}_{\psi\xi} := \{(\psi_i, \xi_i)\}_{i=1}^M$ which are used to approximate the underlying input-output relationship learned by the model. For a new data sample $\psi \in \Psi$, the model trained on $\mathcal{D}_{\psi\xi}$ is used to predict a corresponding target $\xi = \mathfrak{F}_\theta(\psi)$. Recently, there has been a grow-

ing interest in developing data-driven optimization solutions that integrate this learning process with the subsequent optimization process. In this context, one accounts for the fact that the prediction is used within a cost minimization problem $\hat{x}^*(\psi) := \arg\min_{x \in \mathcal{X}} c(x, \mathfrak{F}_\theta(\psi))$, where $\mathcal{X} \subseteq \mathbb{R}^n$ is the set of feasible decisions and $c(x, \xi)$ the cost function. The intent is to adapt the training procedure to produce an adapted decision with low out-of-sample expected cost $\mathbb{E}[c(\hat{x}^*(\psi), \xi)]$.

When there is a mismatch between the training loss $\mathcal{L}$ and the cost function $c(x, \xi)$, a small error in predicting $\xi$ for a given $\psi$ can lead to highly suboptimal $x^*(\psi)$ (see Elmachtoub and Grigas [2022]). Task-based (or decision-focused) learning (c.f. Mandi et al. [2023], Donti et al. [2017]) addresses this issue by training the model $\mathfrak{F}_\theta$ directly on the performance of the policy $x^*(\psi)$. By trading off predictive performance in favor of task performance, the task-based approach can give near optimal decisions.

In high stakes applications, a Decision Maker (DM) usually demonstrates a certain degree of risk aversion by requiring some level of protection against a range of plausible future scenarios. A natural risk averse variant of integrated learning and optimization takes the form of Conditional Robust Optimization (CRO) (see Chenreddy et al. [2022]), which integrates conformal prediction with robust optimization. Specifically, machine learning is first used to estimate an uncertainty set $\mathcal{U}(\psi)$ for an observed context $\psi$. This set $\mathcal{U}(\psi)$, known to contain the realized $\xi$ with a high probability, is then inserted into the conditional robust optimization model:

$$x^*(\psi) := \arg\min_{x \in \mathcal{X}} \max_{\xi \in \mathcal{U}(\psi)} c(x, \xi) \qquad (1)$$

To this date, the methods proposed in the conditional robust optimization literature follow an Estimate Then Optimize (ETO) paradigm. Namely, data is first used to estimate the contextual uncertainty sets which are then calibrated to meet the required coverage levels. These sets are then used as input to the CRO problem to get the adapted robust deci-

sion $x^*(\psi)$. However, the process of calibrating uncertainty sets does not take into account the downstream optimization task, potentially resulting in misalignment between the loss function used in the initial estimation and the objective of robust optimization. In this paper, we propose a novel end-to-end learning framework for conditional robust optimization that constructs the contextual uncertainty set by accounting for the downstream task loss. Our contributions can be described as follows:

- We propose for the first time an end-to-end training algorithm to produce contextual uncertainty sets, $\mathcal{U}(\psi)$ that lead to reduced risk exposure for the solution of the down-stream CRO problem

- We introduce a novel joint loss function aimed at enhancing the conditional coverage of $\mathcal{U}(\psi)$ while improving the CRO objective

- We demonstrate through a set of synthetic environments that our end-to-end approach surpasses ETO approaches at the CRO task while achieving comparable if not superior conditional coverage with its learned contextual set

- We show empirically how our end-to-end learning approach outperforms other state-of-the-art methods on a portfolio optimization problem using real world data from the US stock market

*Remark* 1.1. It is worth noting that when the estimated uncertainty set $\mathcal{U}(\psi)$ reduces to a singleton $\{\mathfrak{F}_\theta(\psi)\}$, i.e. a point prediction, the CRO problem simplifies to the deterministic contextual optimization problem: $x^*(\psi) := \arg\min_{x \in \mathcal{X}} c(x, \mathfrak{F}_\theta(\psi))$. For this special case, the training of $\mathfrak{F}_\theta(\psi)$ using an end-to-end paradigm has been more heavily studied, see for instance Amos and Kolter [2017b], Berthet et al. [2020], Elmachtoub and Grigas [2022]. End-to-end CRO therefore constitutes a more general and unexplored framework that can potentially address the need to provide more robust decisions in situations where parameters cannot be perfectly estimated. This is particularly noticeable in a portfolio optimization problem where a point estimate of the return of assets will necessarily motivate investing all available wealth in the one single asset with highest predicted return. In contrast, it is rather easy to formulate an uncertainty set $\mathcal{U}(\psi)$ such that the CRO problem encourage diversification of the investment.

## 2 RELATED WORK

**Estimate Then Optimize** popularized by the initial work of Hannah et al. [2010] is a framework that integrates machine learning and optimization tasks. Several approaches are proposed to learn the conditional distribution from data. Kannan et al. [2023], Sen and Deng [2018] propose using residuals from the trained regression model to learn conditional distributions. Bertsimas and Kallus [2020] assign weights to the historical observations of the parameters and solve the weighted SAA problem. We refer the readers to the Mišić and Perakis [2020] survey for various applications of the ETO framework. Besides the mentioned risk neutral applications, there is a growing interest in integrating machine learning techniques to Robust Optimization to handle risk-averse scenarios. Chenreddy et al. [2022] identify clusters of the uncertain parameters based on the covariate data and calibrate the sets for these clusters. Patel et al. [2023] propose using non-convex prediction regions to construct uncertainty sets. Blanquero et al. [2023] construct contextual ellipsoidal uncertainty sets by making normality assumptions. Ohmori [2021] use a non-parametric K-nearest neighbors model to identify the minimum volume ellipsoid to be used as an uncertainty set. Sun et al. [2024] solve a robust contextual LP problem where a prediction model is first learned, and then uncertainty is calibrated to match robust objectives. It is to be noted that all these CRO approaches follow the ETO paradigm.

**End-to-end learning** is a more recent stream of work that integrates the Estimation and Optimization tasks and trains using the downstream loss. Donti et al. [2017] proposed using an end-to-end approach for learning probabilistic machine learning models using task loss. Elmachtoub and Grigas [2022] learn contextual point predictor by minimizing the regret associated with implementing prescribed action based on such a point predictor. Amos and Kolter [2017a] use implicit differentiation methods to train an end-to-end model. Butler and Kwon [2023] solve large-scale QPs using the ADMM algorithm that decouples the differentiation procedure for primal and dual variables. Elmachtoub and Grigas [2022] and Mandi et al. [2020] propose using a surrogate loss function to train integrated methods to address loss functions with non-informative gradients. Wang et al. [2023] propose learning a non-contextual uncertainty set by maximizing the expected performance across a set of randomly drawn parameterized robust constrained problems while ensuring guarantees on the probability of constraint satisfaction with respect to the joint distribution over perturbance and robust problems. Costa and Iyengar [2023] propose a distributionally robust end-to-end system that integrates residual based distribution estimation and robustness tuning to the portfolio construction problem. We refer the reader to Kotary et al. [2021], Qi and Shen [2022], Mandi et al. [2023], and Sadana et al. [2023] for broader discussions on both ETO and end-to-end approaches.

**Uncertainty quantification** methods are employed to estimate the confidence of deep neural networks over their predictions (Kontolati et al. [2022]). Common uncertainty quantification approaches include using Bayesian methods like stochastic deep neural networks, ensembling over predictions from several models to suggest intervals, and models

that directly predict uncertain intervals. Gawlikowski et al. [2021]. Beyond estimating predictive uncertainty, ensuring its statistical reliability is crucial for safe decision-making Guo et al. [2017]. Conformal prediction has become popular as a distribution-free calibration method Shafer and Vovk [2008]. Although conformal prediction ensures marginal coverage, attaining conditional coverage in the most general case is desirable Vovk [2012]. Although considered unfeasible, Romano et al. [2020] offers group conditional guarantees for disjoint groups by independently calibrating each group.

## 3 ESTIMATE THEN ROBUST OPTIMIZE

The concept of "Estimate Then Optimize" comes from the contextual optimization literature (see Sadana et al. [2023]). In the context of CRO, the role of the **Estimation** process is to quantify the uncertainty about $\xi$ given the observed $\psi$. This is given as input to an **Optimization** problem that prescribes an optimal contextual decision $x^*(\psi)$.

When the downstream optimization problem is a CRO problem, the estimation step is required to produce a region that adapts to the observed covariates $\psi$ and is expected to contain the response $\xi$ with high confidence. This can be executed in two steps: first, by learning a parametric conditional distributional model denoted as $F_\theta(\psi)$, and second, by calibrating an implied confidence region $\mathcal{U}_\theta(\psi)$ to ensure $\mathbb{P}_{F_\theta(\psi)}(\xi \in \mathcal{U}_\theta(\psi)) = 1 - \epsilon$. For e.g., when one assumes that $\xi|\psi \sim \mathcal{N}(\hat{\mu}(\psi), \hat{\Sigma}(\psi))$, one can learn $(\hat{\mu}(\psi), \hat{\Sigma}(\psi))$ by maximizing the log-likelihood function (see Barratt and Boyd [2023])

$$-\frac{n}{2}\log(2\pi) + \sum_{j=1}^{n} \log L(\psi)_{jj} - \frac{1}{2}\|L(\psi)^\top(\xi - \hat{\nu}(\psi))\|_2^2$$

where $L(\psi)$ and $\hat{\nu}(\psi)$ are the parametric mappings that can be used to compose $\hat{\mu}(\psi) := (L(\psi)^{-1})^\top \nu(\psi)$ and $\hat{\Sigma}(\psi) = (L(\psi)^{-1})^\top L(\psi)^{-1}$. Using the $\alpha$ quantile from the chi-squared distribution with $m$ degrees of freedom, one can define $\mathcal{U}_\theta(\psi)$ that satisfies $\mathbb{P}(\xi \in \mathcal{U}_\theta(\psi)) = 1 - \epsilon$ asymptotically.

Some recent work completely circumvent the need for the intermediary $F_\theta$ by calibrating some $\mathcal{U}_\theta(\psi)$ directly on the dataset. For example, Chenreddy et al. [2022] propose identifying a $k$-class classifier, $a : \mathbb{R}^m \to [K]$ to reduce $\mathcal{U}_\theta(\psi) := \mathcal{U}_\theta(a(\psi))$ such that $\mathbb{P}(\xi \in \mathcal{U}_\theta(k)|a(\psi) = k) \geq 1 - \epsilon \ \forall k$. The literature on conformal prediction also belongs to the family of distribution-free approaches. It separates the calibration of the shape of $\mathcal{U}_\theta(\psi)$ from the calibration of its size, parameterized by a radius $r > 0$, on a reserved validation set to provide out-of-sample marginal coverage guarantees of the form $\mathbb{P}(\xi \in \mathcal{U}_\theta(\psi)) \geq 1 - \epsilon$, where the probability is taken over both the draw of the validation set and of the next sample. According to the Lemma 4.2 in Chenreddy

et al. [2022], such a coverage guarantee is sufficient to ensure that the out-of-sample Value-at-risk of the robust policy produced by CRO is bounded above by the worst-case value of the in-sample problem.

## 4 END-TO-END CONDITIONAL ROBUST OPTIMIZATION

While the ETO approach presented in the section 3 presents an efficient way to quantify the uncertainty conditionally, it does not take into account the quality of the decisions $x^*(\psi)$ that is prescribed by the downstream CRO model. In practice, the quality of a robust decision is usually assessed by measuring the risk associated with the cost produced on a new data sample (a.k.a. out-of-sample). We assume that this risk is measured by a risk measure that reflects the amount of risk aversion experienced by the DM. For instance, one can use conditional value-at-risk represented by the function, $\rho_\alpha(X) := \inf_t t + (1/(1-\alpha))\mathbb{E}[(X-t)^+]$, which computes the expected value in the right tail of the random cost $X$ for a certain risk aversion $\alpha$ and it covers both expected value and the worst-case cost as special cases (i.e. $\alpha = 0$ and 1 respectively).

In the ETO framework, once the optimal decision $x^*(\psi)$ is determined, the DM can assess the associated risk, also known as task loss, $\rho_\alpha(c(x^*(\psi), \xi))$. This metric allows for comparison across models to select the suitable one. However, it is important to note that the model with the best performance in terms of task loss may differ from the optimal model based on prediction loss. Motivated by recent evidence (see Elmachtoub and Grigas [2022]) indicating that performance improvement can be achieved by employing a decision-focused/ task-based learning paradigm, we propose end-to-end conditional robust optimization.

### 4.1 THE ECRO TRAINING PROBLEM

Formally, we let $\Psi \subseteq \mathbb{R}^m$ be an arbitrary support set for $\psi$ whereas $\Xi \subseteq \mathbb{R}^m$ is assumed for simplicity to be contained within a ball centered at 0 of radius $R_\xi$. We consider $c(x, \xi)$ to be convex in $x$ and concave in $\xi$ and let $\mathcal{X}(\psi) := \{x \in \mathbb{R}^n | g(x, \psi) \leq 0, h(x, \psi) = 0\}$ be a convex feasible set for $x$, possibly dependent on $\psi$, and defined through a set of convex inequalities, identified using $g : \mathbb{R}^n \times \mathbb{R}^m \to \mathbb{R}^J$ and affine equalities, identified using an affine mapping $h : \mathbb{R}^n \times \mathbb{R}^m \to \mathbb{R}^J$. The conditional optimal policy then becomes:

$$x^*(\psi, \mathcal{U}) := \arg\min_{x \in \mathcal{X}(\psi)} \max_{\xi \in \mathcal{U}(\psi)} c(x, \xi), \qquad (2)$$

where we make explicit how the decision depends on both the contextual uncertainty set and the realized covariate. Given a parametric family of contextual uncertainty set $\mathcal{U}_\theta$ with $\theta \in \Theta$ and a dataset $D_{\psi\xi} := \{(\psi^i, \xi^i)\}_{i=1}^M$, the ECRO

training problem consists in identifying

$$\min_{\theta \in \Theta} \mathcal{L}_{ECRO}(\theta) := \rho_{i \sim M}(c(x^*(\psi^i, \mathcal{U}_\theta), \xi^i)), \quad (3)$$

where $\rho_{i \sim M}$ refers to the risk when $i$ is drawn uniformly from 1 to $M$, while, for simplicity, we assume $\rho(\cdot)$ to be a conditional value-at-risk measure, and $\mathcal{U}_\theta(\psi)$ to be ellipsoidal for all $\psi$. Namely, we can assume that

$$\mathcal{U}_\theta(\psi) = \mathcal{E}(\mu_\theta(\psi), \Sigma_\theta(\psi), r) \quad (4)$$
$$:= \{ \xi \in \mathbb{R}^m : (\xi - \mu_\theta(\psi))^T \Sigma_\theta(\psi)^{-1}(\xi - \mu_\theta(\psi)) \leq 1 \},$$

for some $\mu_\theta : \mathbb{R}^m \to \mathbb{R}^m$ and $\Sigma_\theta : \mathbb{R}^m \to \mathcal{S}_+$, where $\mathcal{S}_+$ is the set of positive definite matrices, for all $\theta \in \Theta$. While the robust optimization literature suggests various uncertainty set structures that facilitate the resolution of the RO problem, the ellipsoidal set stands out as a natural one to employ as it retains numerical tractability (see Ben-Tal and Nemirovski [1998]) and can easily be described to the DM.

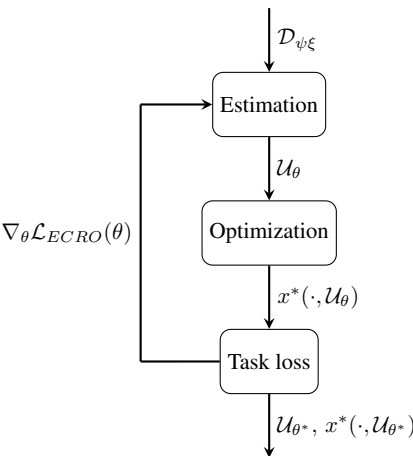

Figure 1: Training pipeline for task-based learning

The training pipeline for the task-based learning approach is illustrated in figure 1. In this pipeline, one starts from an arbitrary $\theta^0$, the optimization problem (2) is solved first for each data point, and the resulting optimal actions are then implemented in order to measure the empirical risk under $D_{\psi\xi}$, which we call empirical ECRO loss of $\theta^0$. A gradient of $\mathcal{L}_{ECRO}(\theta)$ can then be used to update $\theta^0$ in a direction of improvement. Key steps in this pipeline consist in computing $x^*(\psi^i, \mathcal{U}_\theta)$ efficiently and in a way that enables differentiation with respect to $\theta$.

## 4.2 REDUCING AND SOLVING THE ROBUST OPTIMIZATION TASK

Given the convex-concave structure of $c(x, \xi)$ and the convexity and compactness of the ellipsoidal set, we can employ Fenchal duality (see Ben-Tal et al. [2015]) to reformulate the min-max problem as a simpler minimization form over

an augmented decision space. Specifically, we first replace the original cost function with the equivalent cost

$$\bar{c}(x, \xi) := \begin{cases} c(x, \xi) & \text{if } \|\xi\|_2 \leq R_\xi \\ -\infty & \text{otherwise} \end{cases},$$

which integrates information about the domain of $\xi$. One can then employ theorem 6.2 of Ben-Tal et al. [2015], to show that problem (1) can be reformulated as:

$$\min_{x \in \mathcal{X}(\psi), v} f(x, v, \psi) := \delta^*(v|\mathcal{U}_\theta(\psi)) - \bar{c}_*(x, v) \quad (5)$$

where the support function

$$\delta^*(v|\mathcal{U}_\theta(\psi)) := \sup_{\xi \in \mathcal{U}_\theta(\psi)} \xi^T v = \mu_\theta(\psi)^T v + \sqrt{v^T \Sigma_\theta(\psi) v}, \quad (6)$$

while the partial concave conjugate function is defined as

$$\bar{c}_*(x, v) := \inf_\xi v^T \xi - \bar{c}(x, \xi) = \inf_{\xi : \|\xi\|_2 \leq R_\xi} v^T \xi - c(x, \xi).$$

This leads to $x^*(\psi, \mathcal{U}(\psi))$ being the minimizer of the convex minimization problem:

$$\min_{x \in \mathcal{X}(\psi), v} f(x, v, \psi) \quad (7)$$

with $f(x, v, \psi) := \mu_\theta(\psi)^T v + \sqrt{v^T \Sigma_\theta(\psi)} v - \bar{c}_*(x, v)$, a jointly convex function of $x$ and $v$ and finite valued over its domain, and with sub-derivatives:

$$\nabla_v f(x, v, \psi) = \mu_\theta(\psi) + (1/\sqrt{v^T \Sigma_\theta(\psi) v}) \Sigma_\theta(\psi) v - \xi^*(x, v)$$
$$\nabla_x f(x, v, \psi) = \nabla_x c(x, \xi^*(x, v)),$$

where $\xi^*(x, v) := \text{argmin}_{\xi : \|\xi\|_2 \leq R_\xi} v^T \xi - c(x, \xi)$. Revisiting the procedure outlined in figure 1, one can observe that the training process requires a forward pass to find the optimal solutions and a backward pass to update the parameter vector $\theta$. This requires the computation of the gradients of the solution to the problem (3) with respect to the input parameters that are passed through the reformulated CRO problem. Furthermore, the minimization procedure in problem (3) entails navigating through the risk measure $\rho$. These aspects will be further explored in the next section.

## 4.3 GRADIENT FOR PROBLEM (3)

In training problem (3), the gradient of $\mathcal{L}_{ECRO}(\theta)$ with respect to $\theta$ can be obtained using the chain rule:

$$\nabla_\theta \mathcal{L}_{ECRO}(\theta) = \sum_i \frac{\partial \rho_{i \sim M}(y_i)}{\partial y_i}\Big|_{y_i = c(x^*(\psi^i, \mathcal{U}_\theta), \xi^i)} \cdot$$
$$\nabla_x c(\boldsymbol{x})\big|_{x = x^*(\psi^i, \mathcal{U}_\theta)} \cdot$$
$$\left( \nabla_\mu x^*(\psi^i, \mathcal{E}(\mu, \Sigma_\theta(\psi^i)))\big|_{\mu = \mu_\theta(\psi^i)} \nabla_\theta \mu_\theta(\psi^i) \right.$$
$$\left. + \nabla_\Sigma x^*(\psi^i, \mathcal{E}(\mu_\theta(\psi^i), \Sigma))\big|_{\Sigma = \Sigma_\theta(\psi^i)} \nabla_\theta \Sigma_\theta(\psi^i) \right)$$

Based on Ruszczyński and Shapiro, when $\rho(Y) := \text{CVaR}_\alpha(Y)$, one can employ the sub-differential:

$$\nabla_{\boldsymbol{y}} \rho_{i \sim M}(y_i) = \boldsymbol{v}(\boldsymbol{y})$$

with $\boldsymbol{v}(\boldsymbol{y}) \in \text{argmax}_{\boldsymbol{v} \in \mathbb{R}_+^M : \mathbf{1}^T \boldsymbol{v} = 1, \boldsymbol{v} \leq ((1-\alpha)N)^{-1}} \boldsymbol{v}^T \boldsymbol{y}$.

Given that $\nabla_{\boldsymbol{x}} c(\boldsymbol{x})$, $\nabla_\theta \mu_\theta(\psi)$, and $\nabla_\theta \Sigma_\theta(\psi)$ can be readily obtained using auto-differentiation Seeger et al. [2017] when $c(\boldsymbol{x})$, $\mu_\theta(\psi)$, and $\Sigma_\theta(\psi)$ are differentiable, we focus the rest of this subsection on the process of identifying $\nabla_{(\mu, \Sigma)} x^*(\psi, \mathcal{E}(\mu, \Sigma))$. Following the decision-focus learning literature (see Blondel et al. [2022]), one can identify such derivatives by exploiting the fact that any optimal primal-dual pair $(x^*, v^*, \lambda^*, \nu^*)$ of problem (7) must satisfy the Karush-Kuhn-Tucker (KKT) conditions, which take the form:

$$G(x^*, v^*, \lambda^*, \nu^*, \mu, \Sigma, \psi) = 0, \qquad g(x^*, \psi) \leq 0, \lambda^* \geq 0.$$

where

$$G(x^*, v^*, \lambda^*, \nu^*, \mu, \Sigma, \psi) := \\ \begin{bmatrix} \nabla_x f(x^*, v^*, \psi) + \nabla_x g(x^*, \psi)^T \lambda^* + \nabla_x h(x^*, \psi)^T \nu^* \\ \lambda^* \circ g(x^*, \psi) \\ h(x^*, \psi) \end{bmatrix}$$

and $\circ$ denotes the Hadamard product of two vectors.

One can therefore apply implicit differentiation to the constraints $G(x^*, v^*, \lambda^*, \nu^*, \mu, \Sigma, \psi) = 0$ to identify $\nabla_{(\mu, \Sigma)} x^*(\psi, \mathcal{E}(\mu, \Sigma))$ simultaneously with the derivatives of $v^*$, $\lambda^*$, and $\nu^*$ with respect to the pair $(\mu, \Sigma)$. Specifically, one is required to solve the system of equations:

$$\frac{\partial}{\partial x, v, \lambda, \nu} G(x^*, v^*, \lambda^*, \nu^*, \mu, \Sigma, \psi) \cdot \\ \frac{\partial}{\partial (\mu, \Sigma)} (x^*, v^*, \lambda^*, \nu^*)(\mu, \Sigma) = \\ - \frac{\partial}{\partial (\mu, \Sigma)} G(x^*, v^*, \lambda^*, \nu^*, \mu, \Sigma, \psi),$$

where $\frac{\partial}{\partial (x, v, \lambda, \nu)} G$ denotes the Jacobian of the mapping $G$ with respect to $(x, v, \lambda, \nu)$. We refer to Blondel et al. [2022] and Duvenaud et al. [2020] for further details on the computations of related to implicit differentiation.

### 4.4 TASK-BASED SET (TBS) ALGORITHM

In this section, we delve into the implementation details of the ECRO training pipeline. Regarding the contextual ellipsoidal set $\mathcal{E}(\mu_\theta(\psi), \Sigma_\theta(\psi))$, we follow the ideas proposed in Barratt and Boyd [2023] and employ a neural network that maps from $\mathfrak{F}_\theta : \mathbb{R}^m \to \mathbb{R}^m \times \mathbb{R}^{m(m+1)/2} \times \mathbb{R}$. The first set of outputs is used to define $\mu_\theta(\psi)$ while the second and third set forms a lower triangular matrix $L_\theta(\psi)$ and scalar $r_\theta(\psi)$, which is made independent of $\psi$ w.l.o.g., used to

produce $\Sigma_\theta(\psi) := r_\theta(\psi) L_\theta(\psi) L_\theta(\psi)^T$. The positive definiteness of $\Sigma_\theta(\psi)$ is ensured by taking an exponential in the last layer of the network for the output that appears in the diagonal of $L$. The architecture of the neural network can be found in appendix B.6.

The second set of notable details has to do with solving for $x^*(\psi^i, \mathcal{E}(\mu_\theta^i, \Sigma_\theta^i, r_\theta)) \; \forall i$. In our implementation of end-to-end learning for conditional robust optimization, we found that a trust region optimization (TRO) method (see Byrd et al. [2000]) could efficiently solve the reformulated robust optimization problem (7) and provide primal-dual solution pairs for this problem. Given that each episode of the training would pass through the same set of data points, we further observed that the training accelerated significantly (see figure 6 in appendix B.5) when the trust region was interrupted early (after $K = 5$ iterations) as long as it would be warm started at the solution found at the previous epochs. Algorithm 1 presents our proposed training framework for the ECRO approach.

---

**Algorithm 1** ECRO Training with Trust Region Solver

---

1: **input**: dataset $\mathcal{D}_{\psi\xi}$, max epochs $T$, max TRO steps $K$, batch size $N$, protection level $\alpha$
2: Initialize a warm start buffer $\{\bar{x}_1, \ldots, \bar{x}_M\}$ with each $\bar{x}^i \in \mathcal{X}(\psi_i)$
3: Initialize network parameters $\theta$ and $t = 1$
4: **while** not converged and $(t \leq T)$ **do**
5:      Sample a batch of $N$ indices $\mathcal{B} \subset \{1, \ldots, M\}$
6:      **for** $i \in \mathcal{B}$ **do**
7:          //Run TRO for up to $K$ steps
8:          $x_i^t, \lambda_i^t, \nu_i^t \leftarrow \text{TRO}(\bar{x}_i, \mu_\theta(\psi_i), \Sigma_\theta(\psi_i), K)$
9:          $\bar{x}_i \leftarrow x_i^t$          ▷ Update warm start
10:      **end for**
11:      **compute** $\mathcal{L}_{ECRO}(\theta)$ and $\nabla_\theta \mathcal{L}_{ECRO}(\theta)$ for $i \sim \mathcal{B}$
12:      $\theta \leftarrow \theta - \text{step size} \cdot \nabla_\theta \mathcal{L}_{ECRO}(\theta)$
13:      $t \leftarrow t + 1$
14: **end while**
15: **return** $\theta$

---

## 5 END-TO-END CRO WITH CONDITIONAL COVERAGE

Recall that the ETO framework summarized in section 3 focused on producing contextual uncertainty set with appropriate marginal coverage (of $1 - \epsilon$) of the realization of $\xi$. The training pipeline in section 4 was at the other end of the spectrum, disregarding entirely the objective of coverage to increase task performance. In practice, coverage can be a heavy price to pay to obtain performance as it implies a loss in the explainability of the prescribed robust decision. It is becoming apparent that many DMs suffer from algorithm aversion (see [Burton et al., 2020]) and could be reluctant to implement a robust decision produced from an ill covering

uncertainty set.

We further argue that traditional ETO might already face resistance to adoption given the type of coverage property attributed to the ETO sets, i.e. $\mathbb{P}(\xi \in \mathcal{U}(\psi)) = 1 - \epsilon$. Indeed, marginal coverage guarantees only hold in terms of the joint sampling of $\psi$ and $\xi$. This implies that it offers no guarantees regarding the coverage of $\xi$ given the observed $\psi$ for which the decision is made. In fact, a 90% marginal coverage can trivially be achieved if $\mathcal{U}(\psi)$ returns $\Xi$ when $\psi \in \Psi$, for some arbitrary set $\Psi$, and otherwise returns $\emptyset$, as long as $\mathbb{P}(\psi \in \Psi) = 1 - \epsilon$. This is clearly an issue for applications with critical safety considerations and motivates seeking conditional coverage in addition to the marginal coverage when designing $\mathcal{U}(\psi)$. In this section, we outline a training procedure that integrates a sub-procedure that enhances the conditional coverage performance.

## 5.1 THE CONDITIONAL COVERAGE TRAINING PROBLEM

We start by briefly formalizing the difference between the two types of coverage in the definition below.

**Definition 5.1.** Given a confidence level $1 - \epsilon$, a contextual uncertainty set mapping $\mathcal{U}(\cdot)$ is said to satisfy **marginal coverage** if $\mathbb{P}(\xi \in \mathcal{U}(\psi)) = 1 - \epsilon$, and to satisfy **conditional coverage** if $\mathbb{P}(\xi \in \mathcal{U}(\psi)|\psi) = 1 - \epsilon$ almost surely.

The following lemma identifies a necessary and sufficient condition for a contextual set to satisfy conditional coverage.

**Lemma 5.2.** *A contextual uncertainty set $\mathcal{U}(\psi)$ satisfies conditional coverage, at confidence $1 - \epsilon$, if and only if*

$$\mathcal{L}_{CC}(\theta) := \mathbb{E}[\,(\mathbb{P}(\xi \in \mathcal{U}(\psi)|\psi) - (1 - \epsilon))^2\,] = 0$$

*Proof.* For any random variable $X$, one can show that :

$$X = 1 - \epsilon \text{ a.s}$$
$$\Rightarrow \mathbb{E}[(X - (1 - \epsilon))^2] = 1 \cdot (1 - \epsilon - (1 - \epsilon))^2 = 0$$

and that, since $y^2 \leq 0 \Leftrightarrow y = 0$,

$$\mathbb{E}[(X - (1 - \epsilon))^2] = 0$$
$$\Rightarrow (X - (1 - \epsilon))^2 = 0 \text{ a.s. } \Rightarrow X = 1 - \epsilon \text{ a.s..}$$

By letting $X := \mathbb{P}(\xi \in \mathcal{U}_\theta(\psi)|\psi)$, we obtain our result. $\square$

Equipped with lemma 5.2, we formulate the "theoretical" conditional coverage training problem as $\min_{\theta \in \Theta} \mathcal{L}_{CC}(\theta)$. Since the true conditional distribution $\mathbb{P}(\xi \in \mathcal{U}_\theta(\psi)|\psi)$ is typically inaccessible to the DM, we propose an approximation that will make $\mathcal{L}_{CC}(\theta)$ practical.

## 5.2 REGRESSION-BASED CONDITIONAL COVERAGE LOSS

Given a set $\mathcal{U}$, one can define a binary random variable $y(\psi, \xi, \mathcal{U}) := \mathbf{1}\{\xi \in \mathcal{U}(\psi)\}$, and rewrite the conditional probability distribution $\mathbb{P}(\xi \in \mathcal{U}(\psi)|\psi)$ as $\mathbb{P}(y(\psi, \xi, \mathcal{U}) = 1|\psi)$. Using the i.i.d sample data in $\mathcal{D}_{\psi\xi}$, one can approximate this conditional probability using a parametric model, i.e. $\mathbb{P}(y(\psi, \xi, \mathcal{U}) = 1|\psi) \approx g_\phi(\psi)$ for some $\phi \in \Phi$. The parameters $\phi$ can be calibrated by minimizing the negative conditional log-likelihood of $\{y(\psi^i, \xi^i, \mathcal{U})\}_{i=1}^M$:

$$\phi^*(\mathcal{U}) := \arg\min_{\phi \in \Phi} -\frac{1}{M}\sum_{i=1}^M \log g_\phi(\psi^i)^{y^i}(1 - g_\phi(\psi^i))^{1-y^i},$$
(8)

where $y_i := y(\psi^i, \xi^i, \mathcal{U})$. Using the parametric approximation $g_{\phi^*(\mathcal{U})}(\psi) \approx \mathbb{P}(\xi \in \mathcal{U}(\psi)|\psi)$ and replacing the unknown true distribution of $(\psi, \xi)$ with the empirical one, we obtain our regression-based conditional coverage loss function

$$\hat{\mathcal{L}}_{CC}(\theta) := \mathbb{E}^{\mathcal{D}_{\psi\xi}}[(g_{\phi^*(\mathcal{U}_\theta)}(\psi) - (1 - \epsilon))^2].$$

The gradient of $\hat{\mathcal{L}}_{CC}(\theta)$ can be obtained using similar decision-focused training methods as employed for $\mathcal{L}_{ECRO}(\theta)$ given that:

$$\nabla_\theta \hat{\mathcal{L}}_{CC} =$$
$$\sum_{i=1}^M 2(g_{\phi^*(\mathcal{U}_\theta)}(\psi^i) - (1-\epsilon))\nabla_\phi g_{\phi^*(\mathcal{U}_\theta)}(\psi^i)\cdot$$
$$\sum_{j=1}^M \partial\phi^*(\mathcal{E}(\mu, \Sigma_\theta(\psi^i)))/\partial y^j \cdot$$
$$\left(\nabla_\mu y^j(\psi^j, \xi^j, \mathcal{E}(\mu, \Sigma_\theta(\psi^j)))\big|_{\mu=\mu_\theta(\psi^j)}\nabla_\theta\mu_\theta(\psi^j)\right.$$
$$\left.+ \nabla_\Sigma y^j(\psi^j, \xi^j, \mathcal{E}(\mu_\theta(\psi^j), \Sigma))\big|_{\Sigma=\Sigma_\theta(\psi^j)}\nabla_\theta\Sigma_\theta(\psi^j)\right),$$

where the main challenges reside again in the step of differentiating through the minimizer of problem (8).

## 5.3 DUAL TASK BASED SET (DTBS) ALGORITHM

We conclude this section with the presentation of our novel integrated algorithm that learns the contextual uncertainty set network $F_\theta$ by incorporating both the risk mitigation and conditional coverage tasks in the training. Indeed our DTbS training algorithm minimizes the following double task loss function that trades off between the two task objectives:

$$\mathcal{L}_{DT}(\theta) = \gamma\mathcal{L}_{ECRO}(\theta) + (1 - \gamma)\hat{\mathcal{L}}_{CC}(\theta). \quad (9)$$

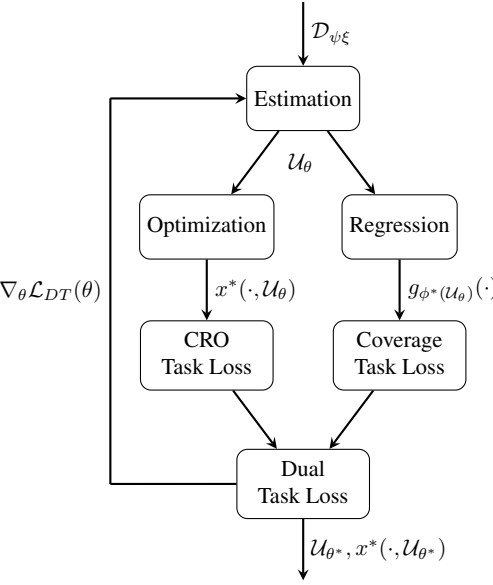

Figure 2: Training pipeline for dual task based learning

The training pipeline for this algorithm can be seen in figure 2. It closely mirrors the structure of the TbS algorithm, with additional crucial steps to compute the necessary components of the loss presented in (9). Within each epoch, the predicted uncertainty set $\mathcal{U}_\theta$ serves two purposes: i) Optimizing CRO to find the optimal policy $x^*(\cdot, \mathcal{U}_\theta)$ and assessing its associated risk; and ii) producing the binary variable $y(\psi, \xi, \mathcal{U}_\theta)$, which regression leading to $g_{\phi^*(\mathcal{U}_\theta)}(\cdot)$ serves to quantify the quality of the conditional coverage. The sum of task losses produces $\mathcal{L}_{DT}(\theta)$, which can be differentiated using decision-focused learning methods. The regression model $g_\phi(\psi)$ takes the form of a feed-forward neural network with a sigmoid activation in the final layer and is optimized using stochastic gradient descent. Algorithm 2 in appendix A presents the details of this DTbS algorithm.

*Remark* 5.3. It is to be noted that achieving distribution-free finite sample conditional coverage guarantees is known to be impossible in the conformal prediction literature (see Barber et al. [2020]). Recently, some progress has been made towards partial forms of conditional coverage guarantees (see Gibbs et al. [2023]) yet it is unclear what are the implications of exploiting such partial coverage properties for the downstream CRO decisions. It is also unclear how such conditional conformal prediction procedures could be integrated within an end-to-end CRO approach.

# 6 EXPERIMENTS

This section outlines our experimental framework devised to demonstrate the advantages of the ECRO method in learning the uncertainty sets tailored to covariate information. Our focus lies in assessing the utility of the model in i) improving the CRO performance; and ii) achieving conditional cover-

age. We conduct a comparative analysis between our two end-to-end approaches, TbS and DTbS, and three state-of-the-art ETO approaches to formulate contextual ellipsoidal sets. We first consider a Distribution-based contextual ellipsoidal uncertainty Set (ETO-DbS) recently introduced in Blanquero et al. [2023], where the conditional distribution of $\xi$ given $\psi$ is presumed to follow a multivariate normal distribution. Additionally, we explore two distributional-free approaches. A vanilla Conformal Prediction Set (ETO-CPS) uses conformal prediction on the output of a point predictor for $\xi$ given $\psi$, after shaping the ellipsoid (through an invariant $\Sigma$) using the residual errors (see Johnstone and Cox [2021]). An Adapted version of Conformal Prediction Set (ETO-ACPS) (proposed in Messoudi et al. [2022]) adapts the shape $\Sigma$ using local averaging around the observed $\psi$. The code can be found on the github[1] repository.

## 6.1 THE PORTFOLIO OPTIMIZATION APPLICATION

We explore the effectiveness of the proposed methodologies in addressing a classic robust portfolio optimization problem. In this context, we define the cost function $c(x, \xi)$ as $-\xi^T x$, where $x$ represents a portfolio comprising investments in $m$ different assets, with their respective returns denoted in the random vector $\xi$. Additionally, we impose constraints on $x$, encapsulated within $\mathcal{X}$, defined as $\mathcal{X} := \{x \in \mathbb{R}^m | \sum_{i=1}^m x_i = 1, x \geq 0\}$. For this cost function, we obtain the partial concave conjugate function:

$$\bar{c}_*(x, v) = \inf_{\xi: \|\xi\|_2 \leq R_\xi} v^T \xi - \xi^T x = -R_\xi \|v - x\|_2 \quad (10)$$

Thus leading to problem (7) becoming

$$\min_{x \in \mathcal{X}} f(x, \psi) := x^T \mu_\theta(\psi) + \sqrt{x^T \Sigma_\theta(\psi) x} \quad (11)$$

when $R_\xi \to \infty$, thus capturing $\Xi := \mathbb{R}^m$.

## 6.2 CRO PERFORMANCE USING SYNTHETIC DATA

We first consider a simple synthetic experiment environment where $m = 2$ and where the pair $(\psi, \xi)$ is drawn from a mixture of three 4-d multivariate normal distributions. We sample $N = 2000$ observations and use 600 observations to train 400 as validation and 1000 observations for testing. All our results present statistics that are based on 10 simulations, each of which employed a slightly modified mixture model (see section B.1 for details). The TbS and DTbS algorithms leverage deep neural networks with the corresponding task losses to learn the necessary components $(\mu_\theta(\psi), \Sigma_\theta(\psi))$ of $\mathcal{U}_\theta(\psi)$. All sets are calibrated for a probability coverage of 90% and the risk of decisions is measured using CVaR

---

[1]https://github.com/Achenred/End-to-end-CRO

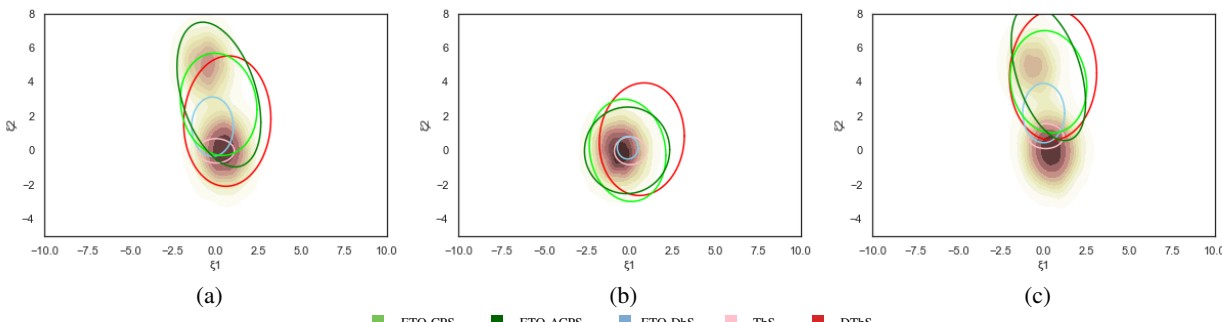

Figure 3: Comparison of uncertainty set ($\alpha = 0.9$) coverage for different $\psi$ realizations: (a) $[2.5, -0.2]^T$, (b) $[-2.6, 0.5]^T$, (c) $[2.7, 1.9]^T$. The shade indicate the true conditional distribution.

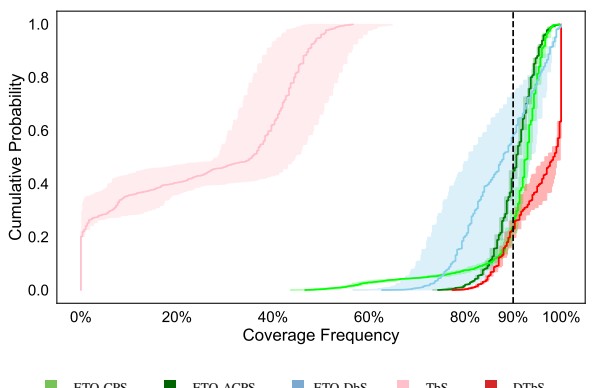

| METHOD | CVaR | MARGINAL COVERAGE |
|---|---|---|
| ETO-CPS | $1.59 \pm 0.03$ | $91 \pm 1.8\%$ |
| ETO-ACPS | $1.68 \pm 0.04$ | $91 \pm 1.4\%$ |
| ETO-DbS | $1.66 \pm 0.06$ | $85 \pm 7.8\%$ |
| TbS | $1.05 \pm 0.09$ | $23 \pm 6.1\%$ |
| DTbS | $1.07 \pm 0.09$ | $92 \pm 1.5\%$ |
| ORACLE | $1.06 \pm 0.10$ | $-$ |

Table 1: Avg. CVaR and marginal coverage for $\alpha = 1 - \epsilon = 0.9$ over 10 simulated environments, error represent 90% CI. Note that the oracle method exploits full information about the Gaussian mixture model.

Figure 4: Average cumulative distribution of conditional coverage frequency when $\psi$ is sampled uniformly from dataset over 10 simulated environments. Shaded region represent 90% CI

at risk level $\alpha = 0.9$. We also consider an "oracle" method that leverages the exact knowledge of the underlying distribution as an additional benchmark. The method is based on formulating a scenario tree approximation of the joint distribution of $\psi$ and $\xi$ in order to obtain an investment policy that minimizes the CVaR objective (3) directly. More details can be found in the Appendix section C. The average CVaR objective values and marginal coverages of the uncertainty sets can be found in the table 1.

One can notice that the end-to-end based methods, TbS and DTbS significantly outperform the ETO methods on the CVaR performance. It appears that in order to maintain the required marginal coverage, the ETO approaches learned sets that resulted in overly conservative RO solutions. We also observe that the TbS and DTbS models achieve a CVaR performance that is very close to our estimate of the best achievable performance, i.e. the oracle method's performance.

Additionally, all the models except TbS appear to have the marginal coverage 90% which corresponds to the $\alpha$ level they are trained for. By disregarding the aspect of coverage,

TbS was able to improve on the CVaR task but performs poorly in terms of coverage. Comparatively, the dual task based approach DTbS was able to improve on the CVaR performance over the ETO approaches while still maintaining the necessary coverage.

As pointed out earlier, conditional coverage is a highly desirable property. Given that a synthetic environment gives us access to exact measurements of conditional coverage, figure 4 presents the cumulative distribution of the observed conditional coverage frequencies when $\psi$ is sampled uniformly from the data set. One can notice from the plot that ETO-DbS, despite being closer to the required marginal coverage, failed to provide accurate conditional coverage. Among the methods that use conformality score to calibrate the radius, ETO-ACPS method which uses localized covariance matrices has better conditional coverage. However, this comes at the price of CVaR performance. The advantages of the dual task-based approach, DTbS, over the single task one are obvious. While DTbS appears to have overshot the coverage compared to ETO-ACPS, which aligns closer to 90%, we argue that this is not an issue as it ends up providing more coverage than needed while generating nearly the best average CVaR value. In figure 3 which overlays the various sets learned on the conditional distribution of $\xi$, one can notice that the sets adapt to the covariate information $\psi$ to provide the necessary conditional coverage.

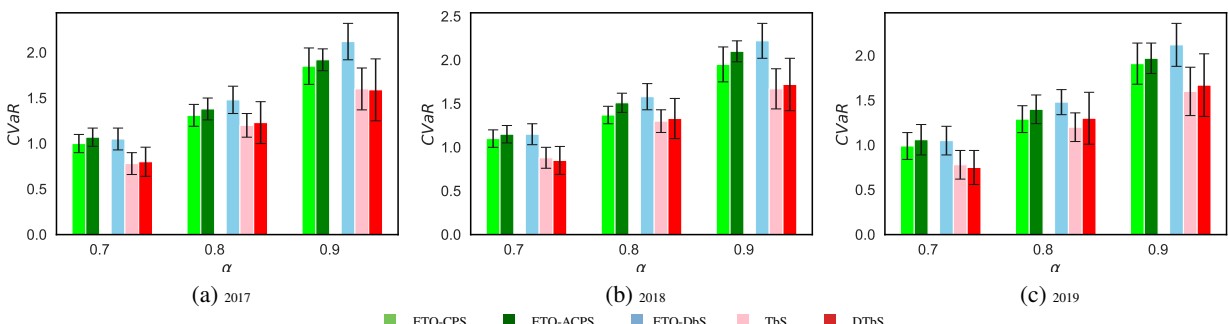

Figure 5: Avg. CVaR of returns across 10 portfolio trajectory simulations. Error bars report 95% CI.

## 6.3 CRO USING US STOCK DATA

We follow the experimental design methodology proposed in Chenreddy et al. [2022]. Our experiments utilize historical US stock market data, comprising adjusted daily closing prices for 70 stocks across 8 economical sectors from January 1, 2012, to December 31, 2020, obtained via Yahoo! Finance's API. Each year contains 252 data points, and we calculate percentage gain/loss relative to the previous day to construct our dataset, denoted as $\xi$. We incorporate the trading volume of individual stocks and other market indices as covariates. We test the robustness of all the model's performance by solving the portfolio optimization problem on randomly selected stock subsets across different periods. Utilizing 15 stocks in each window, we ran the experiment ten times over three moving time frames. We maintain consistent parameters (learning rate $lr$, number of epochs $T$, step size $K$, $\gamma$). Further implementation and parameter tuning details can be found in appendix B.3. Figure 5 compares the avg. CVaR of returns and table 2 presents the marginal coverage across different confidence levels for models.

It is evident from the CVaR comparison that the task based methods TbS and DTbS consistently perform better over the ETO models. Among ECRO approaches, we can clearly observe an advantage for DTbS over TbS, which has on par CVaR performance while having out of sample marginal coverage closer to the expected target level. Conformal-based ETO methods have good marginal coverage as they are designed to have the desired coverage. Especially, ETO-ACPS and ETO-CPS, being calibrated using conformal prediction which produces statistically valid prediction regions have near target coverage levels.

## 7 CONCLUSION

In summary, the paper introduces a novel framework for conditional robust optimization by combining machine learning and optimization techniques in an end-to-end approach. The study focuses on enhancing the conditional coverage of un-

| MODEL | YEAR | MARGINAL COV. (%) | | |
|---|---|---|---|---|
| | | TARGET $1 - \epsilon$ | | |
| | | 70% | 80% | 90% |
| ETO-CPS | | 68 | 78 | 87 |
| ETO-ACPS | | 68 | 77 | 89 |
| ETO-DBS | 2017 | 54 | 72 | 85 |
| TBS | | 22 | 26 | 28 |
| **DTBS** | | **72** | **79** | **88** |
| ETO-CPS | | 67 | 79 | 88 |
| ETO-ACPS | | 68 | 78 | 87 |
| ETO-DBS | 2018 | 59 | 75 | 87 |
| TBS | | 23 | 24 | 29 |
| **DTBS** | | **71** | **80** | **93** |
| ETO-CPS | | 69 | 78 | 88 |
| ETO-ACPS | | 71 | 78 | 89 |
| ETO-DBS | 2019 | 61 | 76 | 86 |
| TBS | | 26 | 30 | 32 |
| **DTBS** | | **69** | **78** | **92** |

Table 2: Marginal Coverage

certainty sets and improving CRO performance. Through comparative analysis and simulated experiments, the proposed methodologies show superior results in robust portfolio optimization. The findings point to the importance of uncertainty quantification and highlight the effectiveness of an end-to-end approach in risk averse decision-making under uncertainty.

### Acknowledgments

The authors gratefully acknowledge support from the Institut de Valorisation des Données (IVADO), the Canadian Natural Sciences and Engineering Research Council [RGPIN-2022-05261], and the Canada Research Chair program [CRC-2018-00105].

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

# A ALGORITHMS

## A.1 DTBS ALGORITHM

---

**Algorithm 2** Dual ECRO Training with Trust Region Solver

---

1: **input**: dataset $\mathcal{D}_{\psi\xi}$, max epochs $T$, max TRO steps $K$, batch size $N$, protection level $\alpha$
2: Initialize a warm start buffer $\{\bar{x}_1, \ldots, \bar{x}_M\}$ with each $\bar{x}_i \in \mathcal{X}(\psi_i)$
3: Initialize network parameters $\theta$ and $t = 1$
4: **while** not converged and $(t \leq T)$ **do**
5:     Sample a batch of $N$ indices $\mathcal{B} \subset \{1, \ldots, M\}$
6:     **for** $i \in \mathcal{B}$ **do**
7:         //Run TRO for up to $K$ steps
8:         $x_i^t, \lambda_i^t, \nu_i^t \leftarrow \text{TRO}(\bar{x}_i, \mu_\theta(\psi_i), \Sigma_\theta(\psi_i), K)$
9:         $\bar{x}_i \leftarrow x_i^t$           ▷ Update warm start
10:         $y_i^t \leftarrow \mathbf{1}\{\xi_i \in \mathcal{E}(\mu_\theta(\psi_i), \Sigma_\theta(\psi_i))\}$
11:     **end for**
12:     $\phi^t \leftarrow$ **solve** prob (8) for $\{(\psi_i, y_i^t)\}_{i\in\mathcal{B}}$
13:     compute $\mathcal{L}_{DT}(\theta)$ and $\nabla_\theta \mathcal{L}_{DT}(\theta)$ for $i \sim \mathcal{B}$
14:     $\theta \leftarrow \theta - \text{step size} \cdot \nabla_\theta \mathcal{L}_{DT}(\theta)$
15: **end while**
16: **return** $\theta$

---

# B SUPPLEMENTARY FOR EXPERIMENTS

## B.1 SYNTHETIC DATA GENERATION PROCESS

Our synthetic experiments rely on a set of mixtures of three multivariate normal distributions created in a way that produces a bimodal mixture of a normal distribution with a possibly non-normal one with similar covariance matrix. Specifically, each mixture model is constructed using the same three mean vectors $\mu_a = \begin{bmatrix} 0 & 0 & 0 & 0 \end{bmatrix}^T$, $\mu_b = \begin{bmatrix} 0 & 5 & 5 & 0 \end{bmatrix}^T$, and $\mu_c = \mu_b$ while the covariance matrices take the form

$$\Sigma_a = \begin{bmatrix} 1 & 0 & 0.37 & 0 \\ 0 & 1.5 & 0 & 0 \\ 0.37 & 0 & 2 & 0.73 \\ 0 & 0 & 0.73 & 3 \end{bmatrix},$$

$\Sigma_b = \alpha\Sigma_a$ and $\Sigma_c = \frac{\Sigma_a}{\alpha}$ for some $\alpha \in [0,1]$, which controls the non-normality of the second mode. Furthermore, we introduce asymmetry in the mixture model by using the mixing proportion $p_a = \phi$, $p_b = \frac{1-\phi}{\alpha+1}$, and $p_c = \frac{\alpha(1-\phi)}{\alpha+1}$ for some $\phi \in [0,1]$, which controls the dominance of the first mode over the second. Furthermore, $p_b$ and $p_c$ are such that the covariance matrix of the non-normal mixture is equal to the covariance of the normal one, $\Sigma_a$.

## B.2 SYNTHETIC CONDITIONAL DATA GENERATION

To generate conditional samples for the synthetic data generated in section B.1, we first compute the conditional mean $\mu_{\xi|\psi}$ and covariance $\Sigma_{\xi|\psi}$ of $\xi$ given the observed variables $\psi$ for each mixture component. Specifically, for each mean vector $\mu$ and covariance matrix $\Sigma$ associated with the mixture components (denoted as $a$, $b$, and $c$ in section B.1), we calculate the conditional parameters as,

$$\mu_{\xi|\psi} = \mu_\xi + \Sigma_{\xi\psi}\Sigma_{\psi\psi}^{-1}(\psi - \mu_\psi)$$

$$\Sigma_{\xi|\psi} = \Sigma_{\xi\xi} - \Sigma_{\xi\psi}\Sigma_{\psi\psi}^{-1}\Sigma_{\psi\xi}$$

Next, we determine the conditional probability of each mixture given the $\psi$ observation using Bayes theorem as $\mathbb{P}(\text{mixture} = i|\psi) \propto \mathbb{P}(\psi|\text{mixture} = i)\mathbb{P}(\text{mixture} = i)$. Finally, we can use these conditional probabilities to sample new data points from the respective conditional distributions of $\xi$ given $\psi$.

## B.3 PARAMETER TUNING PROCEDURE

In this section, we explore the parameter tuning methodology applied to train the network introduced in section 6.3. Given the time series nature of the data, we employ a rolling window technique for network training. Our architecture depends on a set of hyperparameters, defined as follows: $lr$ for learning rate, $T$ for the maximum number of epochs, $K$ for the maximum TRO steps, $B$ for the batch size, and $\alpha$ for the target level. We partition the data into training and validation periods and examine the optimal combination through grid search. For each combination, we train the network and derive the optimal policy using the training data, then apply it to the unseen validation data. The optimal combination is selected based on the lowest CVaR on the validation dataset, viewing this as a worst-case return minimization problem.

Regarding the DTbS algorithm, which balances between two losses—the CRO objective and the conditional coverage loss—we follow a specific strategy to identify the best-performing model. At each epoch, we save the model and initiate model selection only after achieving the required training coverage. Subsequently, we retain the best models meeting the coverage criteria until convergence conditions are met. Among all saved models meeting the coverage requirement, we choose the one with the best CVaR objective.

## B.4 SENSITIVITY ANALYSIS

We conducted a sensitivity analysis of the validation performance as a function of $\gamma$, which balances the CVaR loss and the conditional coverage loss. The table below presents the model performances on the validation data for different

values of $\gamma$. It illustrates how varying $\gamma$ enables a trade-off between the two loss objectives.

| $\gamma$ | 0.01 | 0.1 | 0.5 | 0.9 | 0.99 |
|---|---|---|---|---|---|
| avg. $\mathcal{L}_{\text{ECRO}}$ | 1.30 | 1.05 | 1.04 | 1.06 | 1.05 |
| avg. $\mathcal{L}_{\text{CC}}$ | 5.49 | 6.25 | 8.15 | 8.98 | 8.81 |

## B.5 CONVERGENCE COMPARISON

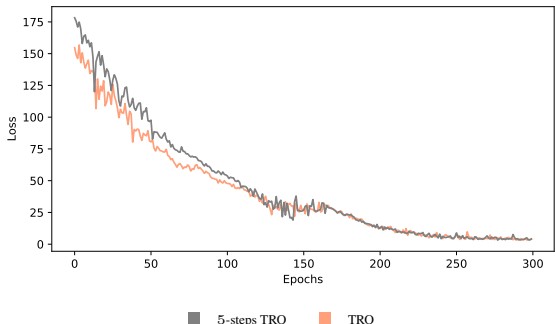

Figure 6: Convergence comparison between 5-steps TRO (46 min) and full TRO (129 min).

## B.6 ARCHITECTURE

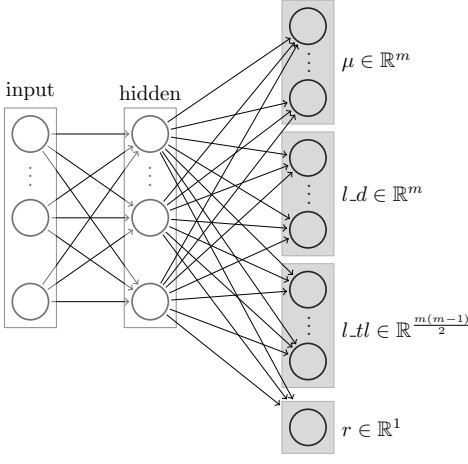

We construct a parametric model for $\mu$ and $\Sigma$ using Cholesky decomposition to ensure positive definiteness of $\Sigma$. We employ a shallow neural network architecture with $m$ input units, one hidden layer of size $h$, and $2m + \frac{m(m-1)}{2} + 1$ units in the output layer. We use tanh for activation functions and softplus for diagonal elements of $L$ to ensure strictly positive values.

## C ORACLE METHOD FOR SYNTHETIC EXPERIMENTS

Given that experiments in section 6.2 are based on a synthetic model, we can evaluate the level of sub-optimality of the portfolio policies proposed by the different models. To do so, we developed an "oracle"-based method that has access to the true underlying joint distribution of $\psi$ and $\xi$ and attempts to identify the "true" optimal value of the CVaR objective, namely

$$\min_{\boldsymbol{x}:\Psi\to\mathcal{X}} \text{CVaR}(-\xi^T x(\psi)).$$

We utilize a scenario tree $\{\psi^i, \{\xi^{ij}\}_{j=1}^M\}_{i=1}^N$ to approximate the joint distribution of $(\psi, \xi)$, where $\psi^i \sim F_\psi$ and $\xi^{ij} \sim F_{\xi|\psi^i}$. Under such scenario tree, the CVaR optimization problem reduces to a linear program:

$$\min_{\{x^i\}_{i=1}^N, \lambda, \{s_{ij}\}_{i=1,j=1}^{N,M}} \lambda + \frac{1}{NM(1-\alpha)} \sum_{i=1}^N \sum_{j=1}^M s_{ij} \quad (12a)$$

$$\text{subject to} \quad s_{ij} \geq 0, \\ \forall i = 1, \dots, N, \, j = 1, \dots, M \quad (12b)$$

$$s_{ij} \geq -(\xi^{ij})^T x^i - \lambda, \\ \forall i = 1, \dots, N, \, j = 1, \dots, M \quad (12c)$$

$$x^i \geq 0, \, \forall i = 1, \dots, N \quad (12d)$$

$$\mathbf{1}^T x^i = 1, \, \forall i = 1, \dots, N. \quad (12e)$$

To be consistent we the test environment, we consider the $\{\psi_i\}_{i=1}^N$, with $N = 1000$, to take on the values of the test set, while $\{\xi^{ij}\}_{j=1}^M$, for each $i$ with $M = 1000$, are randomly sampled from $F_{\xi|\psi^i}$. This is repeated for the 10 problem instances. The average CVaR optimal value of problem (12) is reported in Table 1 as the performance of the oracle method.