# OpenReview forum: "End-to-end Conditional Robust Optimization"
_auai.org/UAI/2024/Conference — UAI 2024 poster_

### Official Review · Reviewer_Ry98 · 2024-03-20

**Q2-1 Originality-Novelty:** 2
**Q2-2 Correctness-Technical Quality:** 3
**Q2-5 Clarity Of Writing:** 3

**Q1 Summary And Contributions:**

This paper is looking at the problem of improving decision making for downstream tasks based on point wise predictions. The authors are motivated by the fact that the majority of the previous works try to solve the prediction and decision making as two separate tasks. Therefore they propose an end to end approach that tries to customize the prediction task to improve the quality of decisions. They back up their arguments and the competency of their approach with both synthetic and real world experiments.

**Q2-3 Extent To Which Claims Are Supported By Evidence:**

3: Good: the main claims are supported by convincing evidence (in the form of adequate experimental evaluation, proofs, (pseudo-)code, references, assumptions).

**Q2-4 Reproducibility:**

3: Good: key resources (e.g. proofs, code, data) are available and key details (e.g. proofs, experimental setup) are sufficiently well-described for competent researchers to confidently reproduce the main results.

**Q3 Main Strengths:**

Their approach is interesting and ambitious. Their method is general enough to be applied in a variety of applications. They have an efficient implementation of their method. This work can potentially bring up insights for some more foundational works on the link between prediction and decision making.

**Q4 Main Weakness:**

The main issue is the lack of more comprehensive discussion on related works. For instance, there is a big bulk of research on the effect of calibrating predictions on the downstream decisions. How do they connect with these line of work?
More fundamentally, it is not entirely obvious why conformal prediction is meaningful here at all. Maybe some other statistics of a prediction set play a more important role for downstream decision makers? If we agree on using CP, how does different CP algorithms (aka different prediction sets) affect your methodology. Furthermore, what is the role of prediction set size in your approach. we know that there is a no free lunch theorem applicable here. One can completely disregard the predictions by the trained ML model and just use non informative, yet valid, prediction sets. How does your approach prevent such a situation? I believe there is a lack foundational thinking behind this work which shows itself in not addressing a number of natural questions.
There is also no major effort to back up the insights\ideas of this paper by a theoretical framework.

**Q5 Detailed Comments To The Authors:**

I suggest to make some changes in the presentation. Specifically, the text doesn't motivate the importance of the role of predictions for the downstream decision makers. Maybe adding a paragraph to illustrate on the matter. The CP shows up out of the blue in this presentation. Maybe some motivation on why CP makes sense here at all. The related work needs more effort to depict a more general picture of different approaches to address this problem.

**Q9 Complying With Reviewing Instructions:**

Yes

---

> ### Author Rebuttal · Authors · 2024-04-06
>
> We appreciate the reviewer for their feedback and suggestions for improvement. We are pleased that he/she finds our paper to be technically sound and are delighted that the reviewer finds our approach interesting and ambitious.
>
> Regarding “Summary and contributions”, we wish to clarify that our contribution is to improve **risk averse** decision making for downstream **robust optimization** tasks based on **set** (rather than point) predictions. Our paper does not offer yet another approach for training point predictors in a way that accounts for downstream task, but rather focus on an emerging field of literature that identifies uncertainty sets used in robust optimization models using machine learning methods.
>
> Regarding “comprehensive discussion of related works”: We agree with the reviewer that the literature review section could benefit from covering more of the literature on decision-focused learning for point prediction. In this regard, we plan on directing the reader to three other relevant surveys [A], [B], and [C]. Conceptually, our approach generalizes this literature to the risk averse setting by allowing the predictor to produce a confidence region (rather than a point) prediction and embedding this region within a robust optimization model. To the best of our knowledge, our paper is the first to perform decision-focused learning this way.
>
> Regarding “why conformal prediction is meaningful here”, which we interpret as “why it is important to predict an uncertainty set”: we argue that uncertainty set prediction is meaningful because we have in mind applications like portfolio optimization where a decision maker (DM) cannot legitimately trust a point prediction embedded within a deterministic optimization model to make their decision. Such DM are usually more comfortable implementing decisions that are solutions to robust optimization models in order to address safety concerns.
>
> Regarding “Maybe some other statistics of a prediction set play a more important role”, we agree with the reviewer. In fact, this constitutes the main motivation for considering end-to-end learning given that some DMs might simply focus on the risk (e.g. CVaR of cost) of the obtained policy. Alternatively, one could consider other statistics than coverage frequency as relevant to the DM. If so, we suspect that our method could be adapted to use as secondary objective a loss that enforces the proper tuning of that statistic.
>
> Regarding “how does different CP algorithms (aka different prediction sets) affect your methodology”, our methodology could easily be adapted to other forms of set predictors as long as the associated support functions are known (see for e.g. Ben-Tal et al. [2015]) for other sets for which one can easily obtain $\delta^*(v|\mathcal{U})$.
>
> Regarding “what is the role of prediction set size”, we agree with the reviewer that one must balance between degree of coverage (large size = larger coverage) and degree of information (smaller = more informed).
>
> “How does your approach prevent such a situation?”: Section 5 explains that our proposed loss functions balances between decision performance and statistical guarantee (see equation (9)). This prevents the sets from becoming non informative when encouraging it to have the right coverage.
>
> Regarding “I believe there is a lack foundational thinking … There is also no major effort to back up the insights\ideas of this paper by a theoretical framework.” We apologize for this omission. Due to lack of space we omitted to summarize the foundational work that had already been established in Chenreddy et al. [2022]. We will include this in the revision. We could also strengthen the theoretical side of the paper by showing that our approach has the potential to perform at least as well as traditional point based decision-focused learning given that the training objective reduces to the point predictor training objective when $\Sigma \rightarrow 0$.
>
> Regarding the detailed comment and “CP shows up out of the blue”, we will add to the paper an illustrative portfolio optimization to motivate the use of an uncertainty set predictor within robust optimization. Indeed, when considering solving $\min_{x\in\mathcal{X}}$ $-\hat{\xi}^T x$ with $\hat{\xi}$ as the output of a point predictor, it is impossible to obtain diversified portfolios using decision-focused learning. This is due to the fact that for all $\hat{\xi}$ the solution will be a vertex of $\mathcal{X}$, which always represents a 100\% investment in a single asset.
>
> [A] Kotary et al., End-to-end constrained optimization learning: A survey. IJCAI 2021.
>
> [B] Qi and Shen, Integrating prediction/estimation and optimization with applications in operations management. Tutorials in Operations Research: Emerging and Impactful Topics in Operations, 36–58, 2022.
>
> [C] Mandi et al., Decision-focused learning: Foundations, state of the art, benchmark and future opportunities. arXiv preprint arXiv:2307.13565.

---

### Official Review · Reviewer_AL7S · 2024-03-22

**Q2-1 Originality-Novelty:** 2
**Q2-2 Correctness-Technical Quality:** 3
**Q2-5 Clarity Of Writing:** 3

**Q1 Summary And Contributions:**

The paper is interested on Conditional Robust Optimization. They propose two end-to-end approach and compare it to the estimate-then-optimize approach.
The first training problem, namely the ECRO minimize the conditional-value-at-risk and the uncertainty set is parametric ellipsoidal set.  Then a gradient for the problem is posed. Finally, an algorithm is proposed using neural network that guaranty the positive of the covariance matrix.
The second training problem focus on the conditional coverage. A necessary and sufficient condition is proposed. Then an approximation is proposed. Finally, an algorithm with convex aggregation of the risk mitigation and the conditional coverage is presented.
To illustrate the effectiveness of end to end approach the portfolio problem is investigated. Frist on synthetic case and then on US stock data.
The results show the interest of the approach and the limitation of the first one which is improve by the second proposal.

**Q2-3 Extent To Which Claims Are Supported By Evidence:**

3: Good: the main claims are supported by convincing evidence (in the form of adequate experimental evaluation, proofs, (pseudo-)code, references, assumptions).

**Q2-4 Reproducibility:**

2: Fair: key resources (e.g. proofs, code, data) are unavailable but key details (e.g. proof sketches, experimental setup) are sufficiently well-described for an expert to confidently reproduce the main results.

**Q3 Main Strengths:**

The paper is clear. The experimentations show the interest of end-to-end approach. More specifically the  and particular the last proposed approach.

**Q4 Main Weakness:**

No sensitive analysis on the parameter \lamba for the DTBS.

**Q5 Detailed Comments To The Authors:**

The DTBS is parametric approach that generalize the ECRO one. It need surprising that a sensitive analysis of the parameter \lambda is not performing and a discussion on how to fix this parameter

**Q9 Complying With Reviewing Instructions:**

Yes

---

> ### Author Rebuttal · Authors · 2024-04-08
>
> We sincerely appreciate the reviewer’s comments and positive view of the paper.
>
> Regarding the sensitivity analysis of the parameter $\lambda$, we omitted to include such a study given that in our implementation of DTbS, the termination criterion of Algorithm 2 is not convergence of $\mathcal{L}_{DT}$ loss (see second paragraph of Appendix B.3 for details). Our termination criterion considers the (in-sample) marginal coverage as a hard constraint while prioritizing the ECRO loss over conditional coverage loss. When experimenting with different values of $\gamma$ (named $\lambda$ in the reviewer’s comment), we observed that the performance of the model obtained from this procedure on the validation set was rather stable for different values of $\gamma$ near 0.5, so decided to simply let it take this default value of 0.5.
>
> For completeness, we include below the table of model performances (on validation data) as a function of $\gamma$. In this table, we can observe that the value of gamma allows to trade-off between the two loss objectives.
>
>
> | $\gamma$ | 0.01  | 0.1  | 0.5  | 0.9  | 0.99  |
> |--------|------|------|------|------|------|
> | avg. $\mathcal{L}_{ECRO}$  | 1.30 | 1.05 | 1.04 | 1.06 | 1.05 |
> | avg. $\mathcal{L}_{CC}$ | 5.49  | 6.25  | 8.15  | 8.98  | 8.81   |
>
> In the revision, we will include a description of our selection process for $\gamma$ and the above sensitivity analysis table.

---

### Official Review · Reviewer_taPD · 2024-03-22

**Q2-1 Originality-Novelty:** 3
**Q2-2 Correctness-Technical Quality:** 4
**Q2-5 Clarity Of Writing:** 4

**Q10 Ethical Concerns:**

None that I could see.

**Q1 Summary And Contributions:**

This paper presents a method for training a decision-focused robust optimization approach. In the proposed setup a ML model outputs an uncertainty, corresponding to a confidence interval for the cost function parameters of an optimization model.

An optimal decision w.r.t. the worst case behavior within the uncertainty set can then be obtained by solving a bi-level problem, which under certain assumptions (mainly convexity w.r.t. the decision variables, concavity w.r.t. the predicted parameters) admits a tractable formulation from which subgradients can be obtained. The model output is then used to compute the decisions-loss, corresponding to the Conditional Value at Risk.

The paper then proposes a tractable formulation for the uncertainty set that is appropriate for continuous parameters; namely, the authors suggest formulating the set as an ellipsoid and calibrating its orientation and scale at training time, so that the level of coverage can be adjusted by still allowing for the flexibility needed to optimize the decision-focused loss.

Finally, the paper discusses how to manage the trade-off between the cost and coverage objectives via a simple convex combination of the two terms.

Empirical results confirm the effectiveness of the approach w.r.t. a classical "predict, then optimize" setup.

**Q2-3 Extent To Which Claims Are Supported By Evidence:**

2: Fair: the main claims are somewhat supported by evidence (but the experimental evaluation may be weak, or does not match entirely with the claims, important baselines may be missing, proofs contain important ideas but lack rigor, algorithmic details are only discussed superficially, references are imprecise, assumptions are not sufficiently motivated or explicated, etc.).

**Q2-4 Reproducibility:**

3: Good: key resources (e.g. proofs, code, data) are available and key details (e.g. proofs, experimental setup) are sufficiently well-described for competent researchers to confidently reproduce the main results.

**Q3 Main Strengths:**

To the best of my knowledge, the paper presents the first complete and viable approach for the end-to-end learning of systems involving prediction and robust optimization. Existing decision focused learning approach can naturally account for uncertainty (under some assumptions), but they do so based on an expected value semantic, and there are real-world settings where controlling the worst case behavior is instead preferable.

The pipeline is well designed and provide a good level of flexibility, while still retaining tractability. The empirical evaluation provides convincing evidence of the methods effectiveness w.r.t. two-stage solutions, on the considered benchmark.

**Q4 Main Weakness:**

Historically, controlling the worst-case behavior is only one of the main motivations for considering robust optimization scenarios. For long time, in fact, robust optimization was often chosen due to its lower computational complexity w.r.t. scenario based approaches for optimization under uncertainty (under properly chosen assumptions).

One of the main appeal of the decision-focused learning emerged in recent years is their ability to account for uncertainty (at least on one-stage problems) without requiring sampling at inference time.
In an end-to-end setup, as this paper shows, robust optimization is in fact significantly harder than classical decision-focused learning, turning one of the historical appeals of the approach into a weakness.

That said, there are definitely real world applications where a worst case semantic is desirable, and for which robust optimization is arguably the recommended solution method.

Along the same lines, while the proposed approach is indeed tractable, the assumptions for its application and the need to solve bi-level optimization problem mostly restrict its direct application to linear cost functions and simple constraints. It is conceivable, however, that approximate solution methods (e.g. subgradient ascent for the predicted parameters) might allow one to tackle more complex problems.

In the empirical evaluation, without a reference value, it is unclear how significant the reported improvements are.

**Q5 Detailed Comments To The Authors:**

* A few terms and acronyms should be defined the first time they are introduced (e.g. the predictive loss, DM, etc.)
* Would it be possible to provide a reference term (e.g. an oracle, or something similar) to provide a better idea of the magnitude of the improvements?
* Do you have any intention to publish the code? That would help a lot with reproducibility

**Q9 Complying With Reviewing Instructions:**

Yes

---

> ### Author Rebuttal · Authors · 2024-04-08
>
> We thank the reviewer for their encouraging words.
>
>
> Regarding “turning the historical appeal of the approach into a weakness”, while we understand the reviewers point of view, we would argue that the appeal of robust optimization is still preserved when comparing to a recent stream of literature interested in applying decision-focused learning to the training a conditional distribution model (see Donti et al. [2017], [A], etc.) that is used in a downstream stochastic program.
>
>
> We also plan to include an illustrative example in the revision that shows the limitation of decision-focused learning with point predictors. Namely, when considering solving $\min_{x\in\mathcal{X}} -\hat{\xi}^T x$ with $\hat{\xi}$ as the output of a point predictor, it is impossible to obtain diversified portfolios using decision-focused learning. This is due to the fact that for all $\hat{\xi}$ the solution will be a vertex of $\mathcal{X}$, which always represents a 100\% investment in a single asset.
>
>
> Regarding “the assumptions for its application … restrict its direct application to linear cost functions and simple constraints”, while we agree that one should expect training to be more demanding for nonlinear objective functions, we wish to emphasize that our assumptions are general enough to allow the application of the method to a broad class of convex minimization problems with cost uncertainty, and convex-concave cost.
>
> Regarding “Would it be possible to include a reference term”: We appreciate the reviewer's suggestion regarding incorporating a reference value, which we agree would enhance the paper. Given the limited timeframe for the rebuttal phase, we have developed an optimal value oracle for the synthetic data experiment (section 6.2) as follows. Utilizing {$\psi^i$,{$\xi^{ij}$}}, with $i=1,...,N$ and $j=1,...,M$, where each $\psi^i\sim F_\psi$ and  $\xi^{ij}\sim F_{\xi|\psi^i}$,  as a scenario tree approximation of for the joint distribution of $(\psi,\xi)$, we solved the CVaR optimization problem:
> $\min_{x:\Psi\rightarrow \mathcal{X}} \mbox{CVaR}_{i\sim N, j\sim M}(-{\xi^{ij}}^T x(\psi^{i})) $
> by reducing it to a linear program (LP). When {$\psi^i$}, with $i=1,...,N$, is the set of out-of-sample data point, this LP provides an “optimal” portfolio policy that exploits knowledge of the underlying distribution and can be used as a benchmark (although not implementable in practice). Furthermore, we also evaluated all models on this scenario tree and compared to the LP optimal value. The table below presents these results (with $M=N=1000$) which will be integrated to the revised version of the paper. In the table, we remark that TbS (and DTbS) actually achieves a CVaR performance that is very close to the best achievable performance (estimated using the scenario tree policy) for the 10 problem instances.
> |              	| Avg. Test CVaR | Avg. Scenario Tree CVaR  |
> |------------------|----------------|-----------------------|
> | ETO-CPS      	| 1.59 $\pm$ 0.03 	| 1.63 $\pm$ 0.04       	|
> | ETO-ACPS     	| 1.68 $\pm$ 0.04	| 1.75 $\pm$ 0.04       	|
> | ETO-DbS      	| 1.66 $\pm$ 0.06	| 1.72 $\pm$ 0.07       	|
> | TbS          	| 1.05 $\pm$ 0.09	| 1.11 $\pm$ 0.08       	|
> | DTbS         	| 1.07 $\pm$ 0.09	| 1.13 $\pm$ 0.08       	|
> | Scenario tree policy | 1.06 $\pm$ 0.1 	| 1.07 $\pm$ 0.09       	|
> (changes in the second decimal are due to models being re-rained during the procedure that tested with the scenario tree benchmark)
> Regarding “Do you have any intention to publish the code?”: yes we do, in fact the submitted paper included a reference to an anonymous [github repository](https://anonymous.4open.science/r/End-to-end-CRO-513E/README.md), where the code can be found.
>
>
> [A] Elmachtoub et al. (2020) Decision trees for decision-making under the predict-then-optimize framework. International Conference on Machine Learning, 2858–2867 (PMLR).

---

### Official Review · Reviewer_3JGh · 2024-03-23

**Q2-1 Originality-Novelty:** 3
**Q2-2 Correctness-Technical Quality:** 3
**Q2-5 Clarity Of Writing:** 2

**Q1 Summary And Contributions:**

The paper introduces an end-to-end framework for Conditional Robust Optimization (CRO), which integrates machine learning and optimization for decision-making under uncertainty. It focuses on the concept of Contextual Optimization, where machine learning predictions are used within a cost minimization problem. The approach addresses the misalignment between predictive loss and cost function in traditional Estimate Then Optimize methods. The framework proposes novel training algorithms to create contextual uncertainty sets that lead to reduced risk in downstream CRO problems.

**Q2-3 Extent To Which Claims Are Supported By Evidence:**

2: Fair: the main claims are somewhat supported by evidence (but the experimental evaluation may be weak, or does not match entirely with the claims, important baselines may be missing, proofs contain important ideas but lack rigor, algorithmic details are only discussed superficially, references are imprecise, assumptions are not sufficiently motivated or explicated, etc.).

**Q2-4 Reproducibility:**

4: Excellent: key resources (e.g. proofs, code, data) are available and key details (e.g. proof sketches, experimental setup) are comprehensively described for competent researchers to confidently and easily reproduce the main results.

**Q3 Main Strengths:**

They Integrated machine learning and optimization in an end-to-end manner for CRO is a novel approach. The framework reduces risk exposure in CRO solutions. The development of a joint loss function that considers both the quality of conditional coverage and CRO performance is interesting. They provides strong empirical evidence using both synthetic data and real-world applications (portfolio optimization using U.S. stock data).

**Q4 Main Weakness:**

No main weakness.

**Q5 Detailed Comments To The Authors:**

I think that a detailed discussion on the computational complexity and scalability of the proposed method would be valuable. Or additional robustness checks, such as sensitivity analysis to key parameters or assumptions, would strengthen the paper.

**Q9 Complying With Reviewing Instructions:**

Yes

---

> ### Author Rebuttal · Authors · 2024-04-08
>
> We thank the reviewer for their encouraging comments on the contributions of our paper. We briefly address their comments below and suggest adding a summary of this discussion to the appendix of the revised paper.
>
> The computational complexity of our algorithm is influenced by key hyperparameters: the learning rate $\alpha$ and the balancing parameter $\gamma$. From our experiments, we observed that using a higher learning rate (eg. 0.1) can sometimes cause the issue of vanishing gradients during training. We found that $\alpha$ around 0.05 to work for a wide range of experiments, which was fixed using cross-validation. We also observed that using a very high value for $\gamma$ (near 0.99) sometimes led to non-convergence and volatile training loss.
>
> Regarding scalability, our $K$-step TRO approach has significantly contributed to reducing the computational complexity of our algorithm, making it suitable for addressing large-scale optimization problems. However, it's worth noting that using too few steps (e.g., $K = 1$) sometimes led to non-convergence issues. However, by employing a reasonable number of steps ($K \geq 5$), we didn't encounter this problem. We finally refer the reviewer to Figure 6 in the appendix for insights on the convergence rate of our $K$-step algorithm.
>
> Finally, regarding robustness, following reviewer AL7S’s comments we ran a sensitivity analysis of the validation performance as a function of $\gamma$, which balances between the CVaR loss and the conditional coverage loss. We include below the table of model performances (on validation data) as a function $\gamma$. In this table, we can observe that the value of gamma allows to trade-off between the two loss objectives.
>
> | $\gamma$ | 0.01  | 0.1  | 0.5  | 0.9  | 0.99  |
> |--------|------|------|------|------|------|
> | avg. $\mathcal{L}_{ECRO}$  | 1.30 | 1.05 | 1.04 | 1.06 | 1.05 |
> | avg. $\mathcal{L}_{CC}$ | 5.49  | 6.25  | 8.15  | 8.98  | 8.81   |
>
> We also added a new benchmark to our comparison of the performance of our methods in the synthetic experiment (section 6.2). This benchmark exploits “optimally” the information about the underlying distribution. While it is not implementable in a real application, it shows us that the performance of TbS and DTbS are nearly optimal for this simple test environment (see response to taPD for more details).

---

### Official Review · Reviewer_CcGD · 2024-03-23

**Q2-1 Originality-Novelty:** 3
**Q2-2 Correctness-Technical Quality:** 2
**Q2-5 Clarity Of Writing:** 2

**Q1 Summary And Contributions:**

In this paper the authors look at the standard Machine Learning setting, and gives an overview of how to find the optimal output.
They argue that "end-to-end conditional robust optimization" is the superior way to find the optimal output.
The paper shows techniques to find this.
In the final section, the authors perform some limited experiments, which show that their method is promising.

**Q2-3 Extent To Which Claims Are Supported By Evidence:**

2: Fair: the main claims are somewhat supported by evidence (but the experimental evaluation may be weak, or does not match entirely with the claims, important baselines may be missing, proofs contain important ideas but lack rigor, algorithmic details are only discussed superficially, references are imprecise, assumptions are not sufficiently motivated or explicated, etc.).

**Q2-4 Reproducibility:**

3: Good: key resources (e.g. proofs, code, data) are available and key details (e.g. proofs, experimental setup) are sufficiently well-described for competent researchers to confidently reproduce the main results.

**Q3 Main Strengths:**

I appreciate the clear problem setting in the introduction. This part clearly indicates what the setting is, and how the authors think of it. This part succeeds well in convincing me that their proposed solution has merits, at the very least in principle.

I also appreciate the fact that the authors include their mathematical derivations in Section 4.

Moreover, I like the fact that the authors included an experiment section.

**Q4 Main Weakness:**

In my mind, this paper has two clear weaknesses.

1. Not all the mathematical objects are defined in the paper. As examples, I think of the notation $\rho_{i\sim M}$ on Page 3, but also the definition of the function $f$ in Eq. (5), where it is not made clear that the right-hand side is not the minimum of $f$, but used to define $f$.
Another instance of this, is $L(\psi)^{-T}$ on Page 2, which I believe is $(L(\psi)^{-1})^{T}$ (this could be standard notation, but I had to look it up).
In general, I find the derivations in Section 4 hard to follow, and I can't make sure whether this is due to the fact that not everything is clearly defined.
The lack of punctuation marks doesn't help here (for instance, how should I understand "when $\psi\in\Psi$ and $\emptyset$ otherwise" on the right column of Page 5?)
I understand that this might not be possible given the page limitations and the theoretical nature of this paper, but for instance Lemma 5.2 is fairly standard and therefore don't require a proof, in my estimation.

2. The language is inconsistent. Some sections read easily, and are with very few (if any) typos. Other sections, such as Section 2, read much more difficult and contain too many typos. This should be improved.

Besides this, the authors do not provide code for their experiments, which renders this section also hard to check or reproduce.

**Q5 Detailed Comments To The Authors:**

Here are my detailed comments.

Abstract, last line: I would put "estimate then optimize" between quotation marks, or indicate with capitals, to increase readability.

Page 1, right column: "When there a mismatch ..." should read "When there is a mismatch ...".
In the same sentence, there is a lacking closing bracket ")".
The next paragraph, DM is not defined, and the word "of" is redundant.

Page 2, Section 2, paragraph "Estimate Then Optimize": The word Popularized should not be capitalized, just as the word "There" near the middle of the paragraph.
There is no space before "Ghenreddy" nor before "Blanquero".

Page 2, Section 2, paragraph "End-to-end learning": "is a more recent stream of work integrates ..." should read "is a more recent stream of work which integrates ..."

Page 3, Section 4, first sentence: "decisions" should be "decision".

Page 3, right column, final equation: Should $R$ not be $R_\xi$ here? The same remark holds throughout the paper, for instance in the next equation.

Page 4, Section 4.4, penultimate sentence: "appear" should be "appears".

Page 7, left column last line: There is a space between = and 2000.

**Q9 Complying With Reviewing Instructions:**

Yes

---

> ### Author Rebuttal · Authors · 2024-04-04
>
> We thank the reviewer for their valuable feedback aimed at enhancing the consistency and readability of our paper. We sincerely apologize for any typos and language inconsistencies that may have affected its readability.
>
> Main weakness #1-2: We can easily address these weaknesses in a revised version of the paper. The paper will be carefully proofread (with a focus on Section 2) and all notation will systematically be introduced. We thank the reviewer for the pointers. In particular, we will explain that $\rho_{i\sim M}(z_i)$ refers to measuring the risk associated to sampling $i$ uniformly from 1 to $M$. We confirm that the suggested interpretation of the other ambiguous notations are accurate.
>
> Regarding “the authors do not provide code for their experiments”, we are surprised to receive this comment as the code was made available on an anonymous [github repository](https://anonymous.4open.science/r/End-to-end-CRO-513E/README.md), which link was provided in section 6.2 of the submitted paper. In the revision, we will make sure to present the url directly in the introduction of section 6 to avoid such oversight.
>
> Given that all the listed weaknesses of the paper can easily be addressed in a minor revision and that the reviewer considers the paper to make non-trivial advances (Q2-1=3), to be technically sound (Q2-2=2), with well supported claims (Q2-3=2), we are surprised to see a final score of 3 attached to this report. We would like to emphasize that our work is the first to seamlessly integrate conditional robust optimization into an end-to-end learning framework, which makes our work a valuable addition to the UAI audience. We would be happy to engage in further discussions with the reviewer to address any other issues that they might identify. Having addressed all the reviewer’s pointers, and considering the effectiveness and validity of our approaches, as highlighted by the other reviewers too, we request the reviewer to reconsider their overall score.

---

### Meta-Review · Area_Chair_6tiD · 2024-04-22

The paper introduces an end-to-end framework for Conditional Robust Optimization (CRO), which combines ideas from machine learning and optimization for decision-making under uncertainty. It focuses on the concept of Contextual Optimization, where machine learning predictions are used within a cost minimization problem.  The authors propose a novel end-to-end approach to train a CRO model in a way that accounts for both the empirical risk of the prescribed decisions and the quality of conditional coverage of the contextual uncertainty set that supports them.

All the reviewers (myself included) are in favor of accepting the paper. The paper provides a novel, interesting contribution to several communities including (deep) learning, uncertainty quantification, and (robust) optimization.